# H3K27me3 and EZH Are Involved in the Control of the Heat-Stress-Elicited Morphological Changes in Diatoms

**DOI:** 10.3390/ijms25158373

**Published:** 2024-07-31

**Authors:** Mhammad Zarif, Ellyn Rousselot, Bruno Jesus, Leïla Tirichine, Céline Duc

**Affiliations:** 1Nantes Université, CNRS, US2B, UMR 6286, F-44000 Nantes, France; mhammad.zarif@univ-nantes.fr (M.Z.); leila.tirichine@univ-nantes.fr (L.T.); 2Institut des Substances et Organismes de la Mer, ISOMer, Nantes Université, UR 2160, F-44000 Nantes, France; bruno.jesus@univ-nantes.fr; 3Institute for Marine and Antarctic Studies (IMAS), Ecology and Biodiversity Centre, University of Tasmania, Hobart, TAS 7004, Australia

**Keywords:** *Phaeodactylum tricornutum*, H3K27me3, EZH, heat stress, photosynthesis, morphotype

## Abstract

Marine water temperatures are increasing due to anthropogenic climate change, constituting a major threat to marine ecosystems. Diatoms are major marine primary producers, and as such, they are subjected to marine heat waves and rising ocean temperatures. Additionally, under low tide, diatoms are regularly exposed to high temperatures. However, physiological and epigenetic responses to long-term exposure to heat stress remain largely unknown in the diatom *Phaeodactylum tricornutum*. In this study, we investigated changes in cell morphology, photosynthesis, and H3K27me3 abundance (an epigenetic mark consisting of the tri-methylation of lysine 27 on histone H3) after moderate and elevated heat stresses. Mutants impaired in PtEZH—the enzyme depositing H3K27me3—presented reduced growth and moderate changes in their PSII quantum capacities. We observed shape changes for the three morphotypes of *P. tricornutum* (fusiform, oval, and triradiate) in response to heat stress. These changes were found to be under the control of PtEZH. Additionally, both moderate and elevated heat stresses modulated the expression of genes encoding proteins involved in photosynthesis. Finally, heat stress elicited a reduction of genome-wide H3K27me3 levels in the various morphotypes. Hence, we provided direct evidence of epigenetic control of the H3K27me3 mark in the responses of *Phaeodactylum tricornutum* to heat stress.

## 1. Introduction

Diatoms are an important group of unicellular phytoplankton, primarily photosynthetic. They contribute to ~20% of the total primary production [1]. Among phytoplankton, the coastal species *Phaeodactylum tricornutum* is distinguished from other diatoms by its frustule, which is either devoid of silica or poorly silicified. The silica deficiency provides this species with cell morphological plasticity. In addition to its morphological flexibility, *P. tricornutum* possesses an extensive array of molecular tools, making it an ideal model to investigate cellular morphological plasticity. This diatom has three major morphotypes: fusiform (FM), oval (OM), and triradiate (TM) [2]. FM is the most abundant and stable morphotype. OM cells are more resistant to stresses such as low salinity or low light, while TM is favored in unstressed conditions [3]. Moreover, several conditions, such as temperature, are known to impact morphotype conversion. However, the impact of high temperatures on marine diatoms as well as the molecular players controlling morphotype conversion, remain mostly unknown. *P. tricornutum* is able to adapt to various environments, and its morphogenesis appears to be independent of its genotype [4,5]. 

*P. tricornutum* is a marine photosynthetic organism using solar photons to produce organic matter, the light energy being converted into chemical energy through electron transfer reactions. The photosystem II, hereafter referred to as PSII, is a multi-protein complex that absorbs light energy. In PSII, there are two antenna proteins, PsbB and PsbC, that transfer energy to the PsbA and PsbD proteins of the reaction center. These four proteins are large transmembrane subunits. At its lumenal side, PSII also contains an oxygene-evolving complex (OEC) made of five proteins (PsbO, PsbQ, PsbU, PsbV, and Psb31) [6]. The OEC performs the water oxidation reaction. Finally, PSII contains small transmembrane proteins such as PsbM [6]. Electron transfer from PSII to PSI is mediated by the plastoquinone and then by the cytochrome *b6-f* complex. This complex is a plastoquinone–plastocyanin reductase. It is composed of several subunits, such as the cytochromes *f* (petA) and *b6* (petB), as well as the petC protein, which is an iron–sulfur cluster protein known as Rieske protein [7]. The plastocyanin electron carrier is absent in *P. tricornutum* and replaced by the cytochrome *c6* [8], which transfers electrons to photosystem I (PSI). In diatoms, PSI is composed of seven trans-membrane subunits (PsaA, PsaB, PsaF, PsaI, PsaJ, PsaL, and PsaM) and three stromal subunits (PsaC, PsaD, and PsaE) [9]. Finally, electrons are transferred from PSI to ferredoxin, which is attached to the PSI through the PsaA-PsaC-PsaE pocket [9]. Diatoms are able to dissipate excess light energy as heat through the binding of xanthophyll pigments to the light-harvesting complex (LHC). Under low light, zeaxanthin, via its antheraxanthin intermediate, is converted to violaxanthin by the zeaxanthin epoxidase (ZEP) enzymes (putatively ZEP1 to 3 in *P. tricornutum*). On the contrary, under excessive light, the reverse reaction is catalyzed by the violaxanthin de-epoxidase (VDE) enzymes (putatively VDE, VDL1, and VDL2 in *P. tricornutum*). A second xanthophyll cycle is constituted by the reversible conversion of diatoxanthin to diadinoxanthin, using the same enzymes [10]. These xanthophyll cycles protect diatoms from high light-induced damage caused to the photosynthetic apparatus. Diatoms are, thus, able to grow under a wide range of light intensities. 

*P. tricornutum* is found in coastal waters, including tidal areas where temperature can be highly variable due to changing water levels. Besides, temperature is known to impact *P. tricornutum* morphology [3]. This diatom is atypical in that it is the only species known to have three major morphotypes. Anthropogenic climate change is notably triggering the warming and acidification of marine waters. These changing conditions impact the growth and physiology of diatoms [11]. The polycomb repressive complex 2 (PRC2) is involved in cell differentiation in various eukaryotes [12]. The PRC2 core proteins are the enhancer of zeste homologs (EZH), suppressor of zeste 12 (SUZ12), embryonic ectoderm development (EED), and retinoblastoma binding proteins 4/7 protein (RBBP4/7, also referred to as p48/p55) [13]. This complex deposits tri-methylation on the lysine 27 of the histone H3 (H3K27me3), an epigenetic mark considered as transcriptionally repressive. In *P. tricornutum*, the homolog of the catalytic subunit of PRC2 is the enhancer of zeste homolog protein (PtEZH). It controls cell differentiation in this species [14]. Previous studies have investigated the role of the H3K27me3 mark in thermo-morphogenesis in *Arabidopsis thaliana*. Indeed, intensive studies reported an epigenetic repression of *FLOWERING LOCUS C* (*FLC*), notably via PRC2 and H3K27me3 [15], and a genome-wide accumulation of H3K27me3 was reported in high temperatures [16]. However, the link between the H3K27me3, its methylase, and the heat-stress morphological responses has never been investigated in *P. tricornutum*.

To date, we know that cell differentiation in *P. tricornutum* is (i) independent of the genotype [2,4], (ii) controlled by PtEZH and H3K27me3 in standard conditions [14], and (iii) modulated by temperature [3]. Hence, we hypothesized that increasing water temperature due to global warming, especially in coastal regions and under low tide, will impact *P. tricornutum* morphology through epigenetic control, such as a modulation of H3K27me3 levels. To test our hypothesis, we used two different heat stress conditions (30 °C and 37 °C) and evaluated diatom responses at the physiological, morphological, molecular, and epigenetic levels. Besides, we used mutants impaired in PtEZH to investigate if and how H3K27me3 controls heat stress response in *P. tricornutum*. Overall, most *Ptezh* mutants have a reduced maximum specific growth rate in optimal growth conditions and moderate changes in their photo-physiological responses. We showed that the three morphotypes of *P. tricornutum* (fusiform, oval, and triradiate) were affected in their cell shape during heat stress (after being grown under standard temperature conditions), a feature lost in *Ptezh* mutants. In response to heat stress conditions (30 °C and 37 °C), several genes encoding proteins involved in photosynthesis were differentially expressed in wild type and *Ptezh* mutants, with different expression patterns. Moreover, there was a strong reduction of genome-wide levels of the H3K27me3 epigenetic mark in response to heat stress.

## 2. Results

### 2.1. Analysis of the Physiological Responses in the Ptezh Mutants under Standard Conditions

*Ptezh* mutant growth was reported to be slightly slower compared to WT [14]. Therefore, we first characterized how *Ptezh* mutants are affected by the absence of H3K27me3 in standard temperature conditions before analyzing the heat stress responses. To do so, we monitored cell growth and photosynthetic features of the *Ptezh* mutants in the FM and TM genetic backgrounds. We used two mutants in each morphotype. FM-*del6*, TM-*M2.10*, and TM-*M2.11* are knock-out mutants [14]. These three mutants produce a truncated version of the PtEZH protein (Appendix A). The fourth mutant, FM-*M2.3*, is a knock-down mutant [14] presenting an insertion of a phenylalanine residue in the EZH1/2-like domain (Appendix A). We monitored cell growth at 19 °C in *Ptezh* mutants for 25 days (Appendix A) and calculated the maximum specific growth rate (µmax). FM-*del6*, FM-*M2.3*, and TM-*M2.11* mutants presented a reduced µmax (Figure 1a,b), while TM-*M2.10* displayed a higher one (Figure 1b). However, TM-*M2.10* and TM-*M2.11* reached a stationary phase at a lower cell concentration compared to TM (Appendix A). Hence, growth is perturbed under normal conditions by the absence of a functional PtEZH protein.

The chloroplast has a central role in the growth of photosynthetic organisms. Since most *Ptezh* mutants exhibited a decreased growth rate, we examined whether the loss of PtEZH could affect photosynthesis in *P. tricornutum*. To do so, we measured several parameters: the α light-limited initial slope for rapid light curves (RLC), the maximal relative electron transport rate (rETRmax), Ek (light saturation coefficient) that corresponds to the photon irradiance at the onset of light saturation coefficient, and maximal non-photochemical quenching (NPQmax) in *Ptezh* mutants. The α values were lower in FM-*M2-3* (Figure 1c) and higher in TM-*M2.11* and TM-*M2.10* (Figure 1d) compared to WT. Only the FM-*del6* mutant displayed a lower photosynthetic performance based on the rETRmax parameter compared to FM (Figure 1e,f). Moreover, the Ek coefficient, which is the light level at which ETR reaches a maximum and the relation between PAR irradiances and ETR stops to be linear, remained unchanged in *Ptezh* mutants compared to WT (Appendix A). Furthermore, NPQmax (Figure 1g,h) and E50 (Figure 1i,j), which is the light level at which 50% of the NPQ occurs, were both reduced in FM-*M2.3* (Figure 1g–i) and TM-*M2.10* (Figure 1h–j) while the other mutants remained unaffected. This suggests that FM-*M2.3* and TM-*M2.10* mutants are less tolerant to higher light intensity.

NPQ enables the dissipation of excessive energy as heat and xanthophylls are important players in this photoprotective mechanism [17]. Since some *Ptezh* mutants had a reduced NPQmax, we investigated transcript levels of the genes encoding enzymes involved in the xanthophyll cycles: (i) the zeaxanthin epoxidase enzymes (ZEP1 to 3) and the violaxanthin de-epoxidase enzymes (VDE, VDL1 and VDL2). Moderate changes in transcript abundance were observed in the *Ptezh* mutants (Figure 2a,b). However, *ZEP1* and *ZEP2* transcript abundance was increased in FM-*del6* and FM-*M2.3*, respectively (Figure 2a). Therefore, the production of enzymes involved in the xanthophyll synthesis pathway did not seem to be altered by the loss of PtEZH.

### 2.2. Gene Expression Analysis of Photosynthesis Proteins in Ptezh Mutants

The measurements described above relied on the fluorescence mainly emitted by PSII. To complete our analysis of *Ptezh* physiology and, more specifically, of photosynthesis in standard conditions, we measured the transcript abundance of various genes encoding proteins involved in photosynthesis in wild type and *Ptezh* mutants. Regarding the PSII complex, transcript abundance of *PsbC*, *PsbA*, and *PsbD* increased in FM-*del6* compared to FM (Figure 3a). Transcript abundance of all PSII tested genes (with the exception of *PsbD*) increased in both TM-*M2.10* and TM-*M2.11* mutants (Figure 3b). Moreover, *petC1* and *Ferredoxin* transcript abundance increased in the four *Ptezh* mutants (Figure 3c). However, a *petC2* increased transcript abundance was monitored in both *Ptezh* TM mutants (Figure 3c). One should note that two genes were identified to encode Rieske proteins, petC1 (Phatr3_J46657) and petC2 (Phatr3_J13358) in *P. tricornutum*. Finally, for PSI, transcript abundance of *PsaA* and *PsaB* was increased in both *Ptezh* FM mutants, while TM-*M2.10* showed an increased *PsaC* transcript level (Figure 3d). Therefore, genes encoding proteins involved in photosynthesis were differentially expressed in *Ptezh* mutants, with different levels of expression in the FM and TM lines (Appendix A).

### 2.3. Morphotype Conversion in Response to Prolonged Heat Stress

To decipher the relationship between temperature and the deposition of H3K27me3 by PtEZH, we applied two types of prolonged heat stress: (i) “moderate heat stress” (MHS) applied for 5 days at 30 °C on 7-day-old cultures pre-acclimated at 19 °C; (ii) “elevated heat stress” (EHS) for 2 days at 37 °C on 5-day-old cultures pre-acclimated at 19 °C.

#### 2.3.1. Morphotype Conversion in Ecotypes of Pure Morphotypes

Temperature impact on morphotypes has previously been assessed in different ecotypes of *P. tricornutum* [3]. However, cultures were analyzed in response to a transition from cold (15 °C) to warm temperature (28 °C) in Pt1 8.6, Pt3, and Pt8 ecotypes. These lines presented a mixture of morphotypes, making it difficult to distinguish between morphotype conversion elicited by heat stress and a differential lethality between morphotypes within the same ecotype. To address this, we purified Pt1 8.6, Pt3, and Pt16 ecotypes to exclusively contain FM (Pt1 8.6-FM), OM (Pt3-OM), or TM (Pt16-TM) cells, respectively, at 19 °C to have representative lines for each morphotype. We used Pt9, a tropical ecotype presenting a mixture of FM and OM cells at 19 °C, as a control. 

In both stress conditions, we observed the appearance of OM cells in Pt1 8.6-FM and FM cells in Pt3-OM (MHS, Figure 4a; EHS, Figure 4c). However, FM cells appeared in Pt16-TM only in response to EHS (Figure 4c). The tropical line Pt9 did not show any changes in morphotype abundance during the first two days in MHS (Figure 4a) and in EHS (Figure 4c). As a control, we assessed the relative abundance of morphotypes for each line maintained at 19 °C and did not observe any changes over time (Appendix A), indicating that morphotype conversion was due to heat stress. We then compared cell growth for each morphotype in order to understand if changes in morphotype abundance could be due to changes in cell viability in response to heat stress. In Pt1 8.6-FM, FM cells display a reduced growth after 2 days at 30 °C (Figure 4b), while OM cells have an increased growth at 30 °C (Figure 4b) and 37 °C (Figure 4d). In Pt3-OM, OM cells displayed a reduced growth after 3 days at 30 °C while FM cells had a constant growth (Figure 4b). In Pt16-TM, TM cells displayed a reduced growth after 2 days at 37 °C (Figure 4d). In Pt9, a similar growth pattern was observed for FM and OM cells in MHS (Figure 4b) and EHS (Figure 4c). Therefore, the changes observed in the relative abundance of morphotypes in response to heat stress were the consequence of decreased growth of the pre-existing morphotype simultaneously with increased growth of the appearing morphotype.

Furthermore, morphotype conversion can occur with or without cell division. To determine whether morphotype conversion observed in response to prolonged heat stress aligns with cell division, we synchronized cells through a prolonged darkness period of 36 h [18]. We then monitored morphotype changes in response to EHS since we noticed changes for the three morphotypes only at 37 °C. We observed that OM cells appeared in Pt1 8.6-FM after 4 h at 37 °C and the abundance is stable after 6 h (Figure 5). FM cells appeared in Pt3-OM after 4 h at 37 °C and then the relative abundance of FM cells were increasing until +8 h (Figure 5). Regarding Pt16-TM, TM cells appeared only after 10 h at 37 °C (Figure 5). *P. tricornutum* divides approximately once per day and cell synchronization decreased after 10 h following illumination [18]. Therefore, morphotype conversion in Pt1 8.6-FM seemed to occur without cell division while, in Pt3, it resulted from OM cells converting to FM cells without cell division and then with cell division. Finally, morphotype conversion in Pt16-TM seemed to occur mainly through cell division.

#### 2.3.2. Regulation of Morphotype Conversion by PtEZH during a Prolonged Heat Stress

We then analyzed how H3K27me3 depletion would impact the heat stress response in *P. tricornutum*. To do so, we monitored the morphotype abundance in various *Ptezh* mutants under both MHS and EHS. *Ptezh* mutants were impaired in PtEZH function and depleted in H3K27me3 [14]. We observed the appearance of OM cells in Pt1 8.6-FM but not in the FM-*del6* and FM-*M2.3* mutants in both MHS (Figure 6a) and EHS (Figure 6c). The TM-*M2.10* and TM-*M2.11* were generated in the Pt8 ecotype [14] presenting a mixture of TM and FM cells [2]. Therefore, we purified Pt8 to have only TM cells (Pt8-TM), as previously carried out for Pt16-TM, the other triradiate ecotype used above. FM cells appeared in the Pt8-TM line in response to EHS (Figure 6d) but not to MHS (Figure 6b). The TM-*M2.10* and TM-*M2.11* mutants have fusiform cells, and no morphotype changes were observed in either MHS (Figure 6b) or EHS (Figure 6d). Therefore, the loss of PtEZH and the associated H3K27me3 depletion led to the loss of temperature-dependent morphotype conversion in all the investigated *Ptezh* mutants. 

### 2.4. Analysis of the Effects of Prolonged Heat Stresses on Photosynthesis Gene Expression

Besides the heat-triggered morphological conversion, we also investigated the physiological response of *Ptezh* mutants in response to heat stress. To do so, we analyzed the transcript abundance of genes encoding (i) enzymes of the xanthophyll cycle, (ii) proteins involved in photosynthesis, and (iii) heat shock proteins. 

Firstly, we investigated the transcript abundance of the genes encoding enzymes involved in the xanthophyll cycles. In response to MHS, the *ZEP1*, *ZEP2*, *ZEP3*, *VDE*, and *VDL1* transcript abundance were reduced in FM and FM-*del6* (Figure 7a). In response to EHS, FM-*M2.3* exhibited increased transcript abundances for all genes (with the exception of *ZEP3*, Figure 7b). On the contrary, FM presented reduced transcript abundances for all genes in EHS (with the exception of *VDL1* and *VDL2*, Figure 7b). The *ZEP2* and *ZEP3* transcript abundance was reduced in FM-*del6* while *VDE* transcript abundance increased in EHS (Figure 7b). Therefore, a prolonged exposure to 30 °C down-regulates the expression of genes encoding enzymes involved in the xanthophyll cycle in the wild type and the FM-*del6* mutant. Surprisingly, a prolonged exposure to 37 °C up-regulates the expression of most genes encoding enzymes involved in the xanthophyll cycle in the knock-down mutant FM-*M2.3*.

Secondly, we measured transcript abundance for a set of photosynthetic genes in FM *Ptezh* mutants subjected to either MHS or EHS. In response to MHS, the *PsbB*, *PsbC*, and *PsbD* transcript abundance increased in FM-*del6* and FM-*M2.3* while the *PsbA*, *PsbO*, *PsbU*, and *PsbM* transcript abundance decreased in wild type and *Ptezh* mutants (Figure 8a). In response to EHS, transcript abundance of most genes related to PSII also decreased (Figure 9a). The *petC2* transcript level decreased in wild type and *Ptezh* mutants under both MHS (Figure 8b) and EHS conditions (Figure 9b). The *Ferredoxin* transcript abundance increased in MHS in wild type and *Ptezh* mutants (Figure 8b) and only in wild type in EHS (Figure 9b). For the PSI complex, *PsaA* and *PsaB* transcript abundances increased in wild type in MHS (Figure 8c) and EHS (Figure 9c) while *PsaC* abundance decreased in both conditions (Figure 8c and Figure 9c). The *Ptezh* mutants exhibited changes in *PsaA*, *PsaB*, and *PsaC* transcript abundance only in response to EHS in (Figure 9c). This suggests that heat stress impacts gene expression of proteins involved in photosynthesis to different extents in wild type and *Ptezh* mutants, depending on the applied temperature.

Thirdly, since heat-shock protein production is modulated by various stresses such as heat, we measured the transcript abundance of a set of genes encoding heat-shock proteins (HSP70G, HSF3, HPSP20, and HPSP20A). Transcript abundances decreased in both wild type and *Ptezh* mutants for the four tested genes in response to MHS (Figure 8d) and EHS (Figure 9d). Therefore, prolonged exposure to increased temperatures greatly decreased the expression of heat-shock proteins.

### 2.5. Analysis of the Effects of Prolonged Heat Stress on H3K27me3 Levels

We showed above that the heat-triggered morphological changes are lost in the absence of a functional PtEZH protein that deposits the H3K27me3 epigenetic mark. Therefore, we investigated whether prolonged heat stress modulates the abundance of H3K27me3 on a genome-wide scale. To do so, we extracted proteins as described by [19] and determined the amount of the H3K27me3 mark by Western blotting under MHS. This experiment was carried out in the Pt1 8.6-FM, Pt3-OM, and Pt16-TM ecotypes, which harbor only FM, OM, or TM cells, respectively, as well as in the Pt9 tropical ecotype. In response to MHS, we observed a complete loss or a strong reduction of H3K27me3 levels in Pt1 8.6-FM, Pt3-OM, and Pt16-TM ecotypes, which represent the FM, OM, and TM morphotypes, respectively (Figure 10). The tropical ecotype Pt9 for which, morphotype abundance remained unchanged in response to MHS (Figure 4a) also exhibited a loss of H3K27me3 (Figure 10). Therefore, exposure to prolonged heat stress leads to a strong reduction of the H3K27me3 epigenetic mark at a genome-wide scale.

## 3. Discussion

In this study, we investigated how some epigenetic players control morphological changes and differentiation in *P. tricornutum*. We hypothesized that PtEZH, the catalytic subunit of the PRC2 complex that deposits the H3K27me3 epigenetic mark, is involved in the heat-elicited morphotype changes. We first carried out a characterization of *Ptezh* mutants to investigate how depletion of H3K27me3 could impact the diatom physiology besides the previously described impact on morphology and differentiation [14]. Second, we applied two prolonged heat stresses (30 °C and 37 °C) to reveal how temperature increase could impact morphological changes and the expression of photosynthesis-related genes in *P. tricornutum* wild type and *Ptezh* mutants. Finally, we investigated how PtEZH could control heat-triggered morphotype changes in *P. tricornutum* and how H3K27me3 levels are modulated by a prolonged heat stress.

In a previous study [14], *Ptezh* mutants showed slower growth monitored over a period of 17 days. Moreover, only FM-*del6* and TM-*M2.11* mutants were analyzed. Here, we monitored the growth for 25 days of the two mutants available in the FM (FM-*del6* and FM-*M2.3*) and TM (TM-*M2.10* and TM-*M2.11*) lines. The maximum specific growth rate was reduced in all mutants, with the exception of TM-*M2.10*. Since TM-*M2.10* showed higher α and lower NPQ values at lower light levels, this mutant might be more efficient in converting light energy under the low light intensities used in our study, which might explain why it grew quicker (Figure 1b) than the other lines. Furthermore, with a longer monitoring time, we observed that only FM-*M2.3*, which is a knock-down mutant, displayed a higher cell concentration at 25 days compared to its wild type (Appendix A). Besides, only FM-*del6* showed a lower photosynthetic performance based on the rETRmax parameter compared to FM. Both FM-*M2.3* and TM-*M2.10* mutants seemed to have an increased sensitivity to excess light energy based on the reduced NPQmax and E50 parameters. Therefore, *Ptezh* mutants show moderate changes in their photosynthetic capacities in standard conditions.

*P. tricornutum* has an optimal growth at 19 °C and a suboptimal growth above 25 °C. This diatom is considered to be eurythermal. To investigate heat stress responses, we chose three purified ecotypes, Pt1 8.6-FM, Pt3-OM, and Pt16-TM, representing the fusiform, oval, and triradiate morphotypes, respectively. The growth rate of OM cells in Pt3-OM was reported to be decreased at 28 °C [3]. We confirmed this finding and found that the conversion of OM cells to FM cells occurs without cell division in the first 4h at 37 °C then in the course of cell division (Figure 5). Besides, the conversion of FM to OM cells in Pt1 8.6-FM at 37 °C occurs without cell division while in Pt16-TM, TM cells convert to FM cells with cell division. In this study, we chose to apply two different temperatures (30 °C–MHS and 37 °C–EHS). The Pt8 ecotype (presenting a mixture of TM and FM cells as described in [3]) did not show any changes in morphotype abundance during a transition from 15 °C to 28 °C [3]. Therefore, we applied a higher temperature to see if we could trigger a heat-elicited morphotype change in a triradiate ecotype. Indeed, both pure triradiate lines used in our study (Pt16-TM and Pt8-TM) showed conversion of TM to FM at 37 °C but not at 30 °C (summarized in Appendix A). It is interesting to notice that the Pt1 8.6-FM, Pt3-OM, and Pt16-TM ecotypes originally contain a mixture of FM/OM, OM/FM, and TM/FM. Thus, heat stress carried out on pure ecotypes (Pt1 8.6-FM, Pt3-OM, and Pt16-TM) triggered the re-appearance of the secondary morphotype. Moreover, this secondary morphotype shows a higher growth rate under heat stress, suggesting its potential role in providing a better adaptation to changing environmental conditions. Given the predominant shift to the fusiform morphotype in response to higher temperatures and its general prevalence, it is plausible that the fusiform morphotype has evolved in response to the rising temperatures experienced over the last century. This morphotype might show an optimized surface-to-volume ratio that confers advantages in challenging environmental conditions and is likely to increase nutrient uptake [20]. Moreover, elongated cells dominate during the summer period [21] when water temperatures are higher, suggesting that higher temperatures favor the fusiform morphotype.

In land plants, photosynthesis is reported to be a highly sensitive physiological process. Indeed, photosynthesis inhibition occurs under moderate heat stress, while the photosynthesis apparatus is damaged under elevated heat stress. More specifically, the photosystem II and its OEC are the most sensitive components [22]. Expression of some genes encoding proteins of petC, PSII, and the light-harvesting complex of the PSI were shown to be down-regulated in warming-adapted populations of *P. tricornutum* [23]. Here, we reported a reduction of *PsbO* and *PsbU* transcript abundance in response to moderate and elevated heat stresses in wild type and *Ptezh* mutants (Appendix A). This likely leads to a reduction in PsbO and PsbU protein levels. These two proteins are cofactors of PSII, and their loss triggers the inactivation of the plant PSII during heat stress [22]. Thus, this strongly suggests a PSII impairment in response to heat stress in *P. tricornutum*. Besides, transcript abundance for genes encoding PSI proteins increased in response to moderate and elevated heat stresses in wild type and *Ptezh* mutants (Appendix A). Hence, *P. tricornutum* might present a higher PSI activity under heat stress. We also observed that transcript abundance for genes encoding cytochrome *b6-f* subunits is affected by prolonged heat stress (Appendix A). Therefore, electron and energy flow from PSII to PSI might follow a different distribution under heat stress compared to optimal growth conditions in the diatom *P. tricornutum*. This is likely associated with a different balance between xanthophyll pigments, based on the changes we observed in the expression of genes encoding enzymes involved in the xanthophyll cycle. Furthermore, the heat-shock proteins participate in cell protection during heat stress and contribute to heat tolerance, notably in plants. It has been recently reported in *P. tricornutum* that HSP70A might be involved in the folding of PsbA and PsbD, the reaction center proteins of PSII [24]. Additionally, *HSP70A* expression was strongly induced after 1 h at 26 °C, although it returned to levels similar to optimal conditions after 5 h [24]. In our study, we observed that prolonged heat stress triggered a decrease in the expression of genes encoding the heat-shock proteins HSP70G, HSF3, HPSP20, and HPSP20A (Appendix A). Therefore, the PSII impairment may result from reduced protection by these molecular chaperones. 

Epigenetic control of heat stress responses has been reported in various eukaryotes (reviewed in [25]). In *P. tricornutum*, specific sets of long non-coding RNAs were identified in each morphotype (FM, OM, or TM), suggesting a role of these epigenetic players in morphotype identity [26]. Furthermore, global DNA methylation levels in CHG and CHH contexts were reported to decrease within gene bodies under warming conditions [23]. Importantly, the triradiate cell shape is abolished in *Ptezh* mutants TM-*M2.10* and TM-*M2.11*, which lack the epigenetic mark H3K27me3 [14]. Our study also reported the role of epigenetics in the control of morphogenesis. Using a reverse genetic approach, we showed that the heat-elicited morphotype changes are lost in the FM-*del6*, FM-*M2.3*, TM-*M2.10*, and TM-*M2.11* mutants (summarized in Appendix A). This led us to conclude that PtEZH controls morphotype changes observed during prolonged heat stresses. It will be interesting to complete this analysis with *Ptezh* mutants in the OM morphotype when available. We also observed a strong decrease or a loss of the H3K27me3 epigenetic mark when 7-day-old cultures of *P. tricornutum*, pre-acclimated at 19 °C, were subjected to prolonged heat stress at 30 °C for 4 days. Thus, *Ptezh* mutants that are already depleted in the H3K27me3 epigenetic mark cannot modulate the levels of this epigenetic mark, leading to an absence of heat-triggered morphotype changes. Therefore, we could postulate that in response to heat stress, PtEZH controls morphological changes via the H3K27me3 epigenetic mark. Further studies would be pertinent in analyzing morphological changes in response to heat stress in mutants impaired in the H3K27me3 demethylase, an enzyme that has not yet been identified [27]. Furthermore, the H3K27me3 levels are more elevated in TM compared to FM [14]. Therefore, a higher temperature might be necessary to elicit the morphotype conversion and a complete H3K27me3 depletion, explaining why morphological changes in triradiate lines Pt8-TM and Pt16-TM were observed at 37 °C, not at 30 °C. Finally, we observed a genome-wide depletion of the H3K27me3 epigenetic mark in response to a moderate heat stress in *P. tricornutum*. However, in *Arabidopsis thaliana*, an opposite observation was reported. Growth under warm temperature (27 °C) led to a genome-wide accumulation of H3K27me3 in this plant [16]. The epigenetic control of heat stress responses might thus be different in marine and terrestrial photosynthetic organisms. Hence, this study opened new avenues regarding the investigation of heat stress responses that are under the control of PtEZH and H3K27me3 in *P. tricornutum*. As further perspectives, it will be relevant to investigate the genome-wide distribution of H3K27me3 and PtEZH in each morphotype (FM, OM, or TM) upon heat stress. In parallel, a transcriptomic analysis will be pertinent to identify key regulators involved in the control of heat-elicited morphotype changes, based on changes in their expression concomitantly with their H3K27me3 levels. Besides, analyzing the cross talk between various epigenetic marks such as the transcription permissive and repressive ones (namely H3K4me3 and H3K36me3 vs. H3K9me3 and H3K27me3) as well as DNA methylation will be valuable to understand how epigenetics triggers heat-elicited morphotype changes in the model diatom *P. tricornutum*.

To conclude, this study reports the first evidence of heat-triggered morphological changes in *P. tricornutum* controlled by PtEZH—the diatom enhancer of zeste homolog that is the catalytic subunit of PRC2—and the epigenetic mark H3K27me3. Each morphotype of *P. tricornutum* adapts differently to a given environment. OM cells, for instance, have better acclimation in benthic environments such as sediment surfaces, showing strong adherence and survival abilities to limiting growth conditions. Conversely, FM and TM cells are better adapted to unstressed growth conditions and pelagic environments [28]. Marine temperature warming will, therefore, modulate morphotype abundance in the diatom *P. tricornutum* and likely in other primary marine producers. This might result in an imbalanced ecosystem impacting community structure and function. 

## 4. Materials and Methods

### 4.1. Used Ecotypes and Genetically-Modified Lines

Several ecotypes of *P. tricornutum* were used in the current study. For the following ecotypes, pure morphotypes were isolated as follows: FM for clone Pt1 8.6 (CCAP 1055/1) originated from an estuary near Blackpool (UK), OM for Pt3 (CCAP 1052/1B, CCMP2558), an ecotype originally isolated from the ecotype Pt2 (Plymouth, UK), TM for Pt16 (RCC641) originated from Helgoland (Helgoland, Atlantic Ocean, North Sea, Germany). To isolate pure morphotypes, cultures presenting a mixture of morphotypes were streaked on EASW-Agar (1/2 EASW [29], 1% agar) plates. After 21 days of culture at 19 °C, under 12/12 light–dark period with a light intensity of 55 μmol photons.m^−2^.s^−1^, isolated colonies were resuspended in EASW. Serial dilutions were performed in 24-well plates to isolate dilutions containing only cells of one single morphotype. All lines were grown axenically using enhanced artificial sea water (EASW) [29] in batch cultures at 19 °C, under 12/12 light–dark period with a light intensity of 55 μmol photons.m^−2^.s^−1^. For the heat stress experiments, we used the tropical ecotype Pt9 (CCMP633) collected in the Territory of Guam (Northern Mariana, Islands, Micronesia), which presents a mixture of fusiform and oval cells at 19 °C [2]. In order to analyze the effects of PtEZH depletion, the following published lines [14] were used: for FM, the Pt1 8.6 ecotype without (referred to as FM-WT) and with the Cas9 control vector introduced by bombardment (referred to as FM-Cas9) and the *Ptezh* knock-out lines FM-Del6 and FM-M2.3; for TM, the Pt8 (CCMP2560) ecotype (described in [2]) collected in the Jericho Beach (Canada) without (referred to as TM-WT) and with the Cas9 control vector introduced by bombardment (referred to as TM-Cas9) and the *Ptezh* knock-out lines TM-M2.10 and TM-M2.11. To assess growth of wild type and *Ptezh* mutants in FM and TM lines, cells were grown in 30 mL of EASW with an initial concentration of 10^5^ cells/mL. Cell counts were measured on triplicate cultures using flow cytometry (CytoFLEX, Beckman Coulter Life Sciences, Brea, CA, USA). Counting was performed every 2 days for 25 days with 1 mL of sample taken from each line. Maximum growth rates (μmax, d^−1^) were determined by fitting growth kinetic data with a Gompertz model [30].

### 4.2. Heat Stress Experiments

The effect of moderated (MHS) and elevated (EHS) heat stress on morphotype conversion and viability was assessed in Pt1 8.6-FM (pure FM), Pt3-OM (pure OM), Pt16-TM (pure TM) and Pt9 (FM + OM) lines. For MHS, cultures were transferred to 30 °C (under 12/12 light–dark period with a light intensity of 55μmol photons.m^−2^.s^−1^) after a 7-day pre-acclimation at 19 °C (under 12/12 light–dark period with a light intensity of 55 μmol photons.m^−2^.s^−1^) of cultures initially inoculated at a concentration of 10^5^ cells.mL^−1^. Then, the cell concentration and abundance of each morphotype were analyzed daily for 5 days with a Malassez chamber for cell counting. Briefly, cells were diluted, and counting was performed on ten chambers, the Malassez cell containing 100 chambers for a total volume of 1µL. For EHS, cultures were transferred to 37 °C (under 12/12 light–dark period with a light intensity of 55 μmol photons.m^−2^.s^−1^) after a 5-day pre-acclimation at 19 °C (under 12/12 light–dark period with a light intensity of 55 μmol photons.m^−2^.s^−1^) of cultures initially inoculated at a concentration of 10^5^ cells.mL^−1^. Then, the cell concentration and abundance of each morphotype were analyzed daily for 2 days with a Malassez chamber for cell counting. For each ecotype, three replicates were monitored and MHS and EHS experiments were repeated three and two times, respectively. 

### 4.3. Heat Stress Experiments on Synchronised Cells

*P. tricornutum* cells were arrested in the G1 phase by prolonged darkness (36 h). Cells were released from this G1 checkpoint by illumination, and they were simultaneously transferred to 37 °C with a light intensity of 55 μmol photons.m^−2^.s^−1^ for heat stress. Cell counting and morphotype abundance were assessed every 2 h during 14 h, starting at illumination (T0). For each ecotype, six replicates were monitored. 

### 4.4. PAM Fluorescence Measurements

For PAM fluorescence measurements, cells were grown at 19 °C (under 12/12 light–dark period with a light intensity of 55 μmol photons.m^−2^.s^−1^) in 20 mL of EASW with an initial concentration of 10^5^ cells/mL. Fluorescence analyses were performed every day and data analyses (presented in Figure 1c–j and Appendix A) were performed when Fv/Fm was at its maximum. The Fv/Fm parameter represents the maximum PSII quantum efficiency and was calculated as follows: Fv/Fm = (Fm − Fo)/Fm, with Fm being the maximum fluorescence yield and Fo the minimum fluorescence yield. Photosynthesis analysis was carried out at room temperature with a pulse amplitude-modulated fluorimeter (Imaging PAM; Walz). Before measurement, samples were dark-adapted for 60 min. Thirteen increasing PAR irradiances (from dark to 633 μmol photons.m^−2^.s^−1^) were applied for 30 s with a blue measuring light (450 nm), controlled by the software ImagingWin v2.46i (Heinz Walz GmbH, Effeltrich, Germany) to generate the rapid light curves (RLC), measures being carried out in a dark room. The following parameters were inferred from RLC data: α is the light-limited initial slope under light limitation, Ek (light saturation coefficient) is the photon irradiance at the onset of light saturation coefficient, the PSII maximum relative electron transfer rate (rETRmax, relative unit), the maximum non-photochemical quenching (NPQ_max_) and the E50 that is the light level at which 50% of the NPQ occurs. These parameters were calculated as described in [4]. Six replicates were used for each genotype. The RLC rETR data were fitted by the function defined in [31], and the RLC NPQ data were fitted by the function in [32].

### 4.5. Protein Extraction and Immunoblot Analysis

Proteins were recovered from chromatin extracted for each ecotype as previously described in [19] but without formaldehyde fixation of cell cultures. Protein concentration was assessed in each sample thanks to a BCA protein assay (Pierce™ BCA Protein Assay Kit, Thermo Scientific™, Waltham, MA, USA), using a Bovine Serum Albumin as a standard range. Proteins were resolved on Mini-PROTEAN TGX Stain-Free Gels (4–15% precast polyacrylamide gels, Biorad, Hercules, CA, USA) and then electroblotted onto Trans-Blot Turbo Transfer Pack PVDF (Biorad) with a Trans-Blot Turbo Transfer System (Biorad). Immunoblots were probed with the anti-H3K27me3 antibody (Cell Signalling Technologies, Danvers, MA, USA; C36B11, lots GR242835-1, GR265016-1, and GR172700-1; 1/3000). Equal loading was confirmed with an anti-H4 antibody (Merck Millipore, MA, USA; 05-858, lot 3836545; 1/1000). Primary antibodies were revealed by incubation with an anti-rabbit (Promega, Madison, NY, USA; W4018, lot 0000417842; 1/1000) secondary antibody. Immunoblot chemiluminescence was revealed using Clarity Max Western ECL substrate detection reagents (Biorad, Hercules, CA, USA). Imaging was performed with a Chemidoc™ Imaging System (Biorad, Hercules, CA, USA).

### 4.6. RNA Extraction and Quantitative RT-PCR

Total RNAs were extracted with 1 mL of TRIzol® Reagent (Invitrogen, Thermo Fisher Scientific, USA) according to manufacturer instructions, followed by a DNase I (NEB) treatment and a purification through phenol-chloroform extraction. The cDNAs were synthesized with ImProm-II™ Reverse Transcription System (Promega) from 2 µg of total RNAs and random hexamers, according to manufacturer instructions. The cDNAs were then diluted four times. For qPCR, 1 µL cDNA was used with the Takyon No ROX SYBR 2X MasterMix blue dTTP (Eurogentec) according to the manufacturer’s instructions on a Biorad Cycler. Transcript abundance was calculated as follows: Relative Transcript Level = 2^−(Ct_gene of interest-Ct_RPS)^. For *petC1*, *petC2*, and *Ferredoxin*, the following formula was used: Relative Transcript Level = 10^3^ − 2^−(Ct_gene of interest-Ct_RPS)^. The *RPS* (ribosomal protein small subunit 30S) gene was used for normalization [33]. Two genes were identified to encode Rieske proteins, petC1 (Phatr3_J46657), and petC2 (Phatr3_J13358). RT-qPCR histograms show mean amplifications ± SE of three biological replicates and two technical qPCR replicates. Primers used in this study are listed in Appendix A. Before use, the efficiency of each primer set was assessed. To do so, it was calculated with the following formula: E = 10^(−1/−slope)^, the slope being determined from the standard curve obtained with serial dilutions of cDNAs.

### 4.7. Protein Sequence Alignments and Structures

The protein sequences were aligned with the Clustal Omega (1.2.4) program [34]. The 3D protein modeling was performed with the Phyre2 (2.0) [35] web portal and superimposed with the Chimera software v1.16 [36]. Protein domains were determined from InterProScan (5.69-101.0) [37].

## Figures and Tables

**Figure 1 ijms-25-08373-f001:**
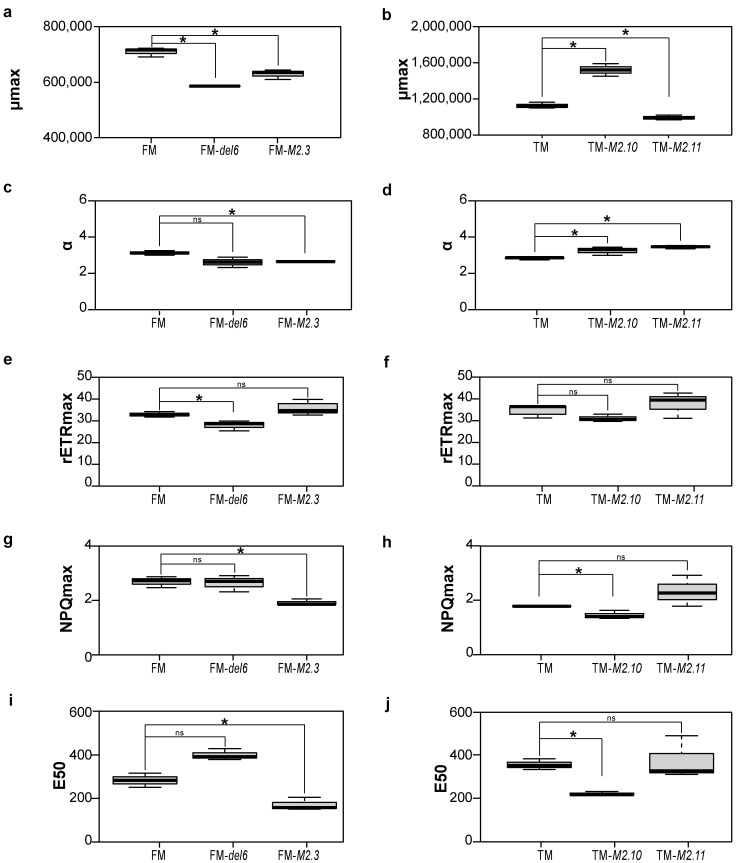
Growth and photosynthesis analysis of *Ptezh* mutants in FM and TM lines. (**a**,**b**) Growth rates are represented with error bars indicating standard deviations based on triplicate cultures in FM (**a**) and TM (**b**) lines. Maximum growth rates were determined from the growth curve obtained by cell counting performed every 2 days for 25 days on cultures grown at 19 °C (12/12 light–dark period; light intensity of 55 μmol photons.m^−2^.s^−1^). (**c**–**j**) Photosynthesis analysis performed when Fv/Fm was at its maximum for the *Ptezh* mutants in FM and TM lines grown at 19 °C (12/12 light–dark period; light intensity of 55 μmol photons.m^−2^.s^−1^). (**c**,**d**) α slope for rapid light curves in FM (**c**) and TM (**d**) lines. (**e**,**f**) rETRmax is the PSII maximum relative electron transfer rate in FM (**e**) and TM (**f**) lines. (**g**,**h**) NPQmax is the maximum non-photochemical quenching in FM (**g**) and TM (**h**) lines. (**i**,**j**) E50 is the light level at which 50% of the NPQ occurs in FM (**i**) and TM (**j**) lines. Student’s *t*-test: * *p* < 0.05; ns, non-significant.

**Figure 2 ijms-25-08373-f002:**
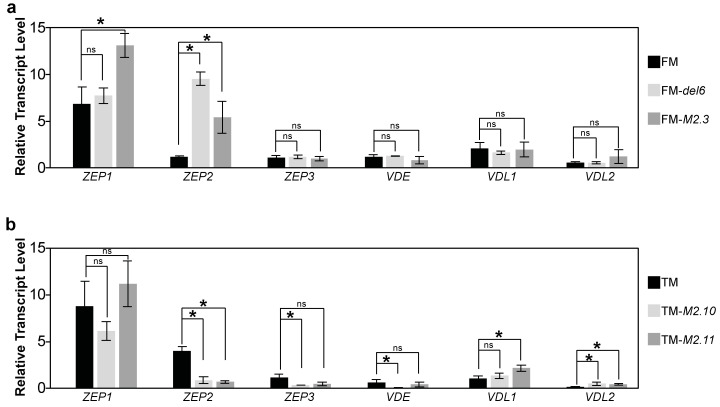
Expression of genes encoding zeaxanthin epoxidase (ZEP), violaxanthin de-epoxidase (VDE), and violaxanthin de-epoxidase-like (VDL). (**a**,**b**) Relative transcript levels of genes encoding ZEP1, ZEP2, ZEP3, VDE, VDL1, and VDL2 enzymes in the FM (**a**) and TM (**b**) lines. The ZEP1, ZEP2, and ZEP3 enzymes putatively convert (i) zeaxanthin via the antheraxanthin intermediate to violaxanthin and (ii) diadinoxanthin in diatoxanthin. The VDE, VDL1, and VDL2 enzymes putatively catalyze the reverse reactions [10]. Levels are measured by qRT-PCR on three biological replicates consisting of 5-day-old cultures at 19 °C (12/12 light–dark period; light intensity of 55μmol photons.m^−2^.s^−1^) in (**a**,**b**). Student’s *t*-test: * *p* < 0.05; ns, non-significant.

**Figure 3 ijms-25-08373-f003:**
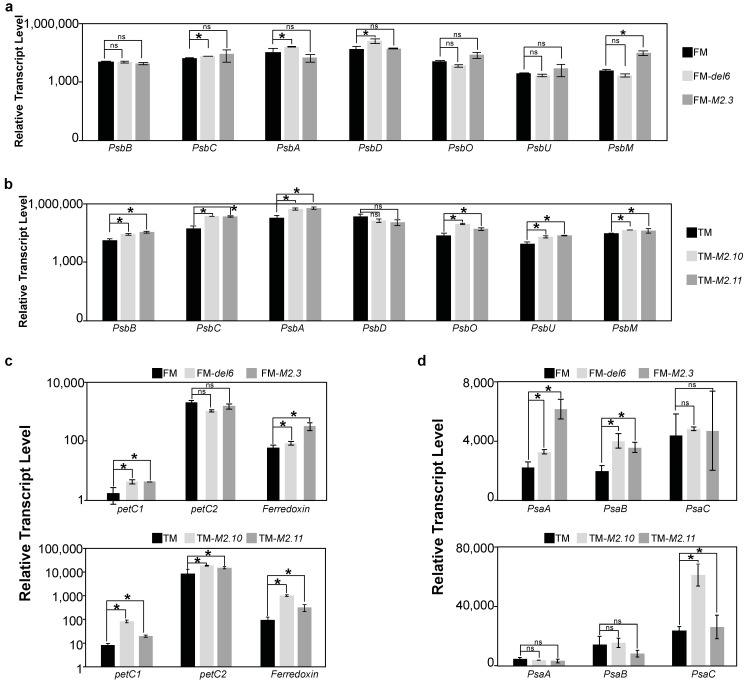
Expression of genes encoding proteins of the photosynthesis apparatus. (**a**,**b**) Relative transcript levels of genes encoding proteins of PSII (antenna: PsbB and PsbC; reaction center: PsbA and PsbD; OEC: PsbO and PsbU; small transmembrane protein: PsbM) in the FM (**a**) and TM (**b**) lines. (**c**) Relative transcript levels of genes encoding proteins of the cytochrome b6-f complex (Rieske proteins: petC1 and petC2) and ferredoxin in the FM (upper panel) and TM (lower panel) lines. (**d**) Relative transcript levels of genes encoding proteins of PSI (trans-membrane subunits: PsaA, PsaB; stromal subunit: PsaC) in the FM (upper panel) and TM (lower panel) lines. Levels are displayed with a logarithmic scale (**a**–**c**) and measured by qRT-PCR on three biological replicates consisting of 5-day-old cultures grown at 19 °C (12/12 light–dark period; light intensity of 55 μmol photons.m^−2^.s^−1^) in (**a**–**d**). Student’s *t*-test: * *p* < 0.05; ns, non-significant.

**Figure 4 ijms-25-08373-f004:**
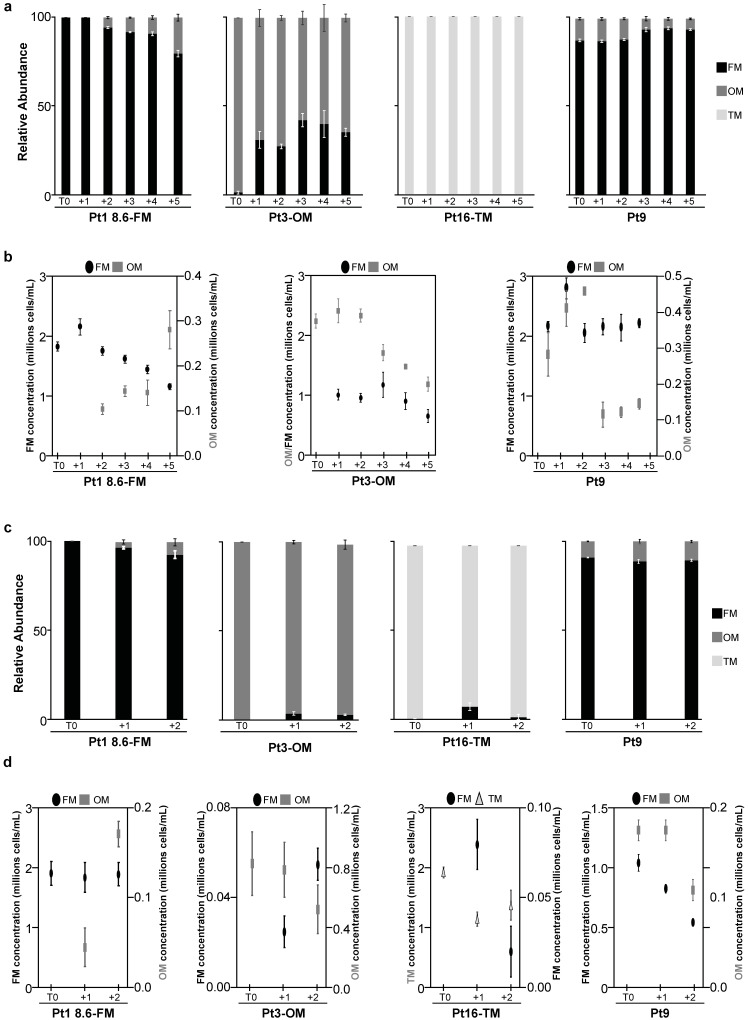
The impact of prolonged-heat stress exposures in representative ecotypes of *P. tricornutum*. (**a**,**c**) Relative abundance of each morphotype in Pt1 8.6-FM, Pt3-OM, and Pt16-TM that are representative of the FM, OM, and TM morphotypes, respectively. The Pt9 tropical strain was used as a control. (**b**,**d**) Cell concentration measured for each morphotype (FM and OM) for Pt1 8.6-FM, Pt3-OM and Pt9 subjected to MHS (**b**) and for Pt1 8.6-FM, Pt3-OM, Pt16-TM, and Pt9 subjected to EHS (**d**). Relative abundance presented in panels *a* and *c* were inferred from these measures. A moderate heat stress (MHS) was applied for 5 days at 30 °C to 7-day-old cultures pre-acclimated at 19 °C (**a**,**b**). An elevated heat stress (EHS) was applied for 2 days at 37 °C to 5-day-old cultures pre-acclimated at 19 °C (**c**,**d**). Measures were carried out before transferring the cultures to heat-stress conditions (T0). They were then performed every day for 5 days in MHS and 2 days in EHS after transfer under heat stress.

**Figure 5 ijms-25-08373-f005:**
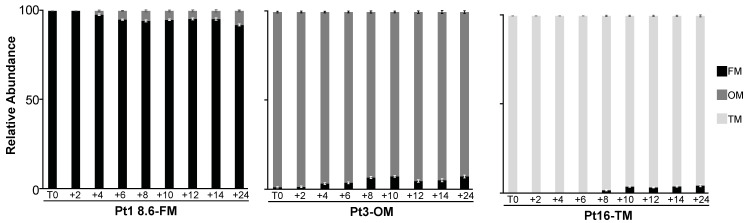
Kinetics of the morphological changes observed during exposure of 5-day-old cultures to 37 °C in representative ecotypes of *P. tricornutum*. Synchronized cells were transferred to 37 °C at T0, simultaneously with illumination to release cell cycle progression from G1. The relative abundance of each morphotype in Pt1 8.6-FM (**left**), Pt3-OM (**middle**), and Pt16-TM (**right**) was then assessed every 2 h for 14 h.

**Figure 6 ijms-25-08373-f006:**
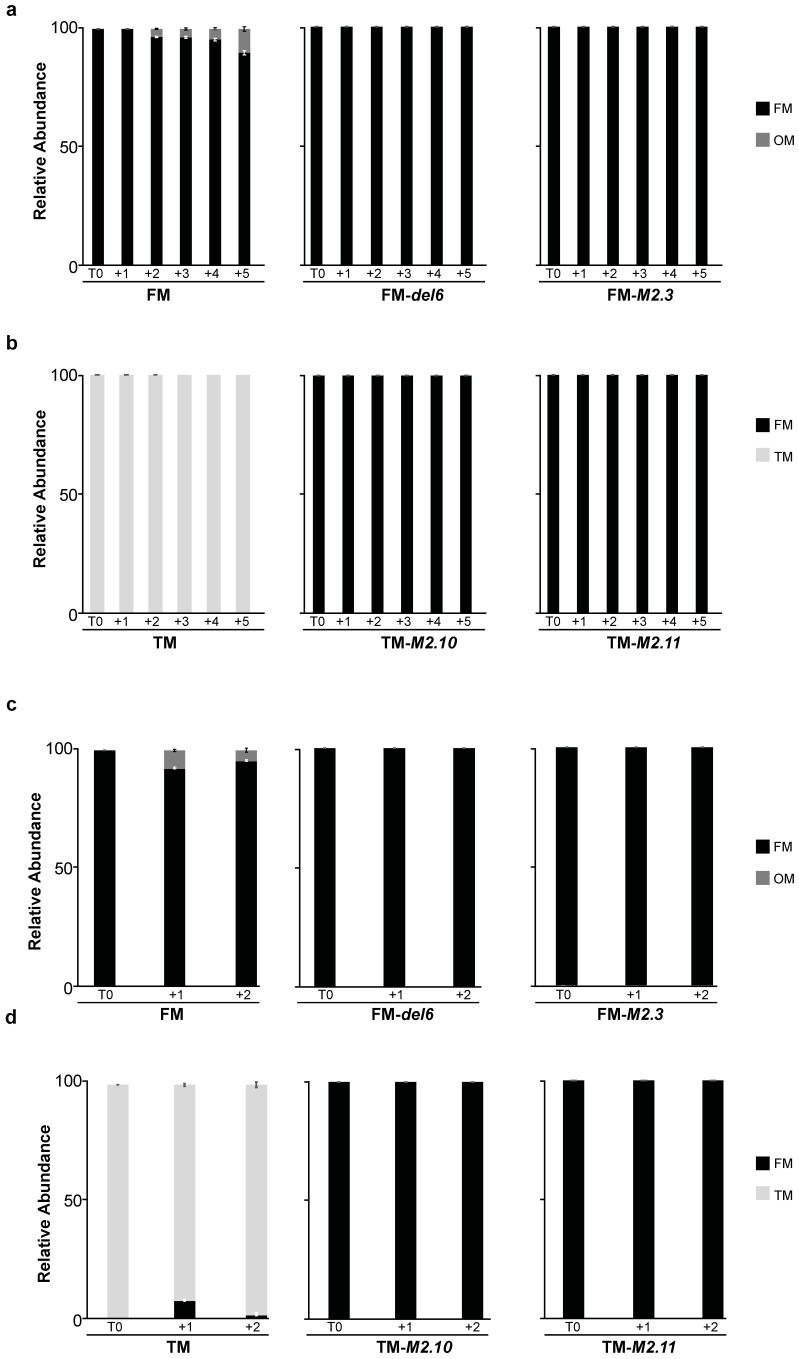
Impact of prolonged-heat stress exposures on the morphology of *Ptezh* mutants. (**a**,**b**) Relative abundance of FM (**a**) and TM (**b**) lines subjected to a moderate heat stress (MHS). (**c**,**d**) Relative abundance of FM (**c**) and TM (**d**) lines subjected to an elevated heat stress (EHS). A moderate heat stress (MHS) was applied for 5 days at 30 °C on 7-day-old cultures pre-acclimated at 19 °C (**a**,**b**). An elevated heat stress (EHS) was applied for 2 days at 37 °C on 5-day-old cultures pre-acclimated at 19 °C (**c**,**d**). Measures were carried out before transferring cultures to heat stress conditions (T0). They were then performed every day for 5 days in MHS and 2 days in EHS after transfer under heat stress.

**Figure 7 ijms-25-08373-f007:**
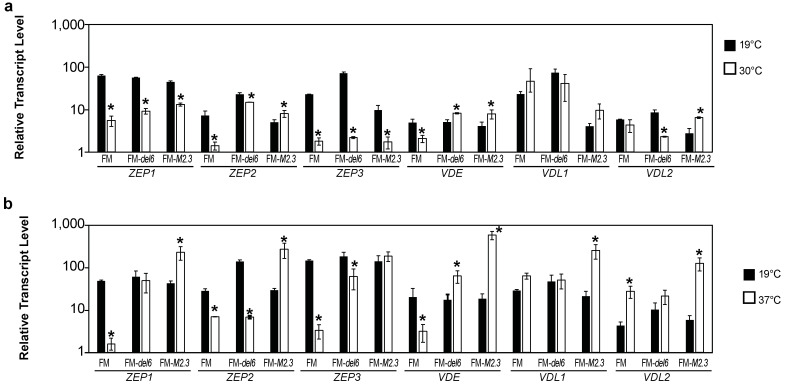
Expression of genes encoding zeaxanthin epoxidase (ZEP), violaxanthin de-epoxidase (VDE), and violaxanthin de-epoxidase-like (VDL) in response to prolonged heat stresses. (**a**,**b**) Relative transcript level of genes encoding ZEP1, ZEP2, ZEP3, VDE, VDL1, and VDL2 enzymes in MHS (**a**) and EHS (**b**) lines. The ZEP1, ZEP2, and ZEP3 enzymes putatively convert (i) zeaxanthin *via* the antheraxanthin intermediate to violaxanthin and (ii) diadinoxanthin in diatoxanthin. The VDE, VDL1, and VDL2 enzymes putatively catalyze the reverse reactions [10]. Levels are displayed with a logarithmic scale and measured by qRT-PCR on three biological replicates consisting of 7-day-old cultures pre-acclimated at 19 °C transferred to 30 °C for 4 days (**a**) and 5-day-old cultures pre-acclimated at 19 °C transferred to 37 °C for 2 days (**b**). Student’s *t*-test; * *p* < 0.05.

**Figure 8 ijms-25-08373-f008:**
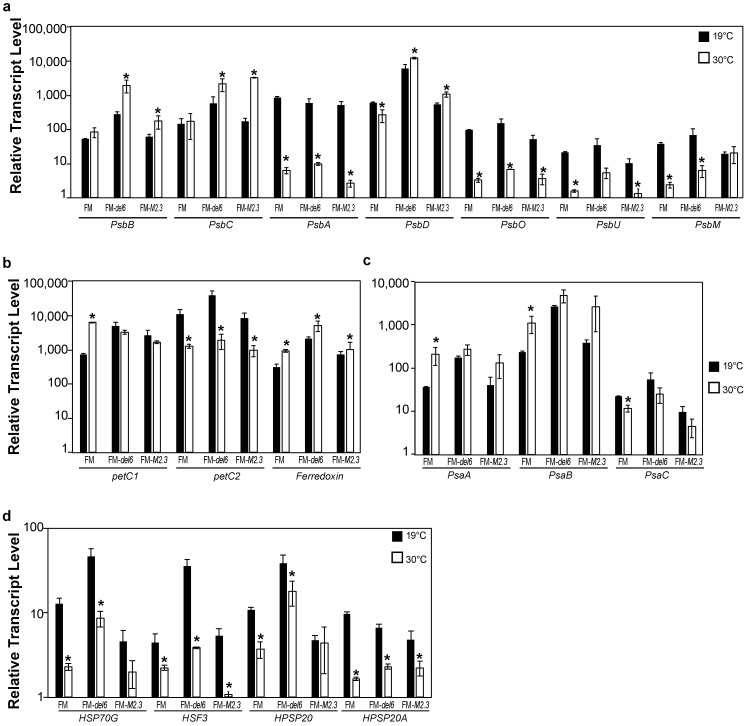
Expression of genes encoding proteins of the photosynthesis apparatus in response to MHS. (**a**) Relative transcript levels of genes encoding proteins of PSII (antenna: PsbB and PsbC; reaction center: PsbA and PsbD; OEC: PsbO and PsbU; small transmembrane protein: PsbM). (**b**) Relative transcript levels of genes encoding proteins of the cytochrome b6-f complex (Rieske proteins: petC1 and petC2) and ferredoxin. (**c**) Relative transcript levels of genes encoding proteins of PSI (trans-membrane subunits: PsaA, PsaB; stromal subunit: PsaC). (**d**) Relative transcript levels of genes encoding heat shock proteins (HSP, heat shock protein; HSF3, heat shock factor protein 3). Levels are displayed with a logarithmic scale and measured by qRT-PCR on three biological replicates consisting of 7-day-old cultures pre-acclimated at 19 °C transferred to 30 °C for 4 days. Student’s *t*-test; * *p* < 0.05.

**Figure 9 ijms-25-08373-f009:**
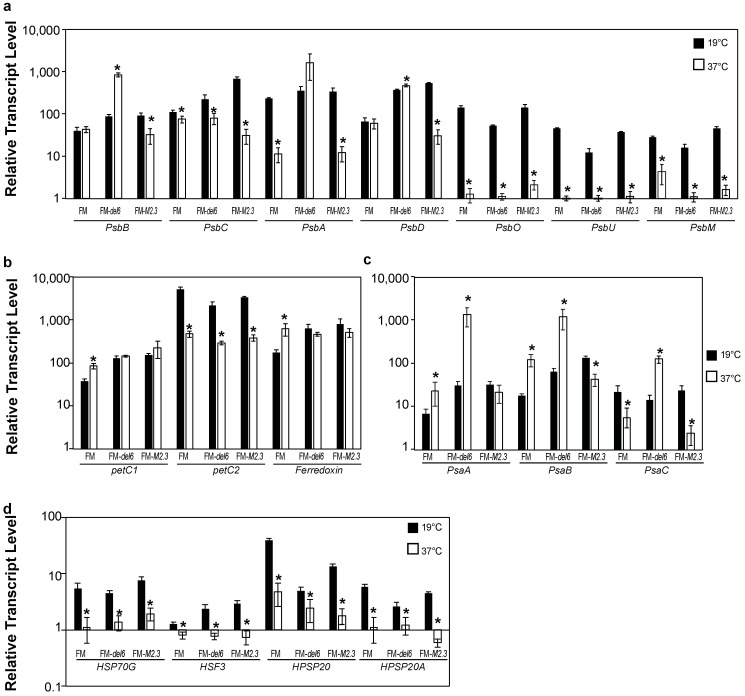
Expression of genes encoding proteins of the photosynthesis apparatus in response to EHS. (**a**) Relative transcript level of genes encoding proteins of PSII (antenna: PsbB and PsbC; reaction center: PsbA and PsbD; OEC: PsbO and PsbU; small transmembrane protein: PsbM). (**b**) Relative transcript level of genes encoding proteins of the cytochrome b6-f complex (Rieske proteins: petC1 and petC2) and ferredoxin. (**c**) Relative transcript level of genes encoding proteins of PSI (trans-membrane subunits: PsaA, PsaB; stromal subunit: PsaC). (**d**) Relative transcript level of genes encoding heat shock proteins (HSP, heat shock protein; HSF3, heat shock factor protein 3). Levels are displayed with a logarithmic scale and measured by qRT-PCR on three biological replicates consisting of 5-day-old cultures pre-acclimated at 19 °C transferred to 37 °C for 2 days. Student’s *t*-test; * *p* < 0.05.

**Figure 10 ijms-25-08373-f010:**
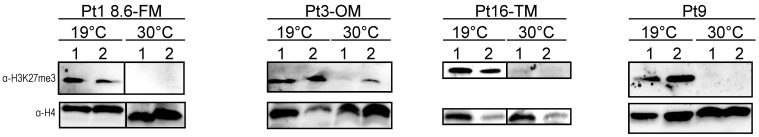
Genome-wide levels of the epigenetic mark H3K27me3 in response to MHS. The H3K27me3 levels were quantified by Western blotting in 7-day-old cultures pre-acclimated at 19 °C transferred to 30 °C for 4 days. H4 was used as a loading control. Presented blots are representative of several blots and biological replicates. Two biological replicates (referred to as 1 and 2) consisting of independent cultures collected at the same time are presented for each ecotype.

## Data Availability

All data are included in this article.

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
