# Peer review of "H3K27me3 and EZH Are Involved in the Control of the Heat-Stress-Elicited Morphological Changes in Diatoms"

_ijms, 2024, doi:10.3390/ijms25158373_

Round 1

Reviewer 1 Report

Comments and Suggestions for Authors

The manuscript describes a series of experiments carried out on various strains (clans) of Phaeodactylum tricornutum collected from an estuary near Blackpool (UK) and isolated from ecotype Pt2 (Plymouth, UK), and from Heligoland (Heligoland, Atlantic Ocean, North Sea, Germany ), pure tested morphotypes were isolated from these ecotypes. They were subjected to two long-term heat stresses (30°C and 37°C) to discover how an increase in temperature might affect morphological changes and the expression of photosynthesis-related genes in wild-type P. tricornutum and Ptezh mutants. Changes in cell morphology, photosynthesis process and H3K27me3 content were analyzed, which allowed to demonstrate the epigenetic control of this mark in the reactions of the diatom P. tricornutum to heat stress. The article is interesting and worth publishing with some corrections.

Line 25: In light of the statement that “in Arabidopsis thaliana, an opposite observation was reported. Growth under warm temperature (27°C) led to a genome-wide accumulation of H3K27me3 in this plant [16]. The epigenetic control of heat stress responses might thus be different in marine and terrestrial photosynthetic organisms.” (lines 486-489). The statement “Hence, we provided direct evidence of epigenetic control of the H3K27me3 mark in diatom responses to heat stress” should be modified. Most likely, the research results cannot be applied to all diatoms. This probably applies only to species living in the sea, and certainly applies to Phaeodactylum tricornutum.

Line 74-76: Does shape change affect many (if not all) diatom species, or just Phaeodactylum tricornutum? I suggest clarifying this sentence.

Lines 70-75, The paragraph ending with the sentence "However, the impact of high temperature on marine diatoms remains mostly unknown, as well as the molecular players controlling morphotype conversion.", has nothing to do with the substantive content later in this paragraph (from the sentence: “The Polycomb Repressive Complex 2 (PRC2) is involved in cell differentiation in various eukaryotes [12].”) – I propose two paragraphs. I suggest moving the first one closer to the beginning of the "Introduction" section.

Lines 83 and 99; 109, 135 and 136: Why is it sometimes written " Ptezh " and elsewhere " PtEZH"?

Lines 100-102: “We showed that the three morphotypes of P. tricornutum (fusiform, oval and triradiate) were affected in 101 their cell shape during heat stress …” - This has been shown previously [3], this study showed “only” , that this feature is lost in Ptezh mutants.

Line 119: According to Fig. 1a and 1b, only TM-M2.10 had a higher µmax value, and not as in the sentence "…. while TM-M2.11 displayed a higher one (Fig. 1b)”.

Lines 225-226 and Figure 4: "The tropical line Pt9 did not show any changes in morphotype abundance in either MHS (Fig. 4a) ..." However, according to Figure 4a, between the 3rd and 4th day of line Pt9 exposure on temperature, the number of OM cells decreased by a similar amount as in the Pt1 8.6-FM morphotype.

Lines 438-439 and 461: "(Wan et al. 2023)" – not in the literature list.

Author Response

Point-by-Point Response to Reviewer 1 comments:

  1. Summary: We would like to thanks both reviewers who provided detailed edits. We addressed them all. These comments greatly improved our manuscript entitled "H3K27me3 and EZH are involved in the control of the heat-stress-elicited morphological changes in diatoms" by Zarif et al. (ijms-3104800). We hope that these answers will satisfy both reviewers and the Editor and sufficiently improve our manuscript for publication in IJMS. Please find the detailed responses below and the corresponding corrections highlighted in red in the re-submitted files.

2. Questions for General Evaluation

Reviewer’s Evaluation

Response and Revisions

Does the introduction provide sufficient background and include all relevant references?

Can be improved

Addressed in Comment 1 and 2 and 3

Are all the cited references relevant to the research?

Can be improved

Addressed in Comment 8

Is the research design appropriate?

Can be improved

Addressed in Comment 2

Are the methods adequately described?

Yes

More details were added upon request of reviewer 2

Are the results clearly presented?

Can be improved

Addressed in Comments 6 and 7

Are the conclusions supported by the results?

Yes

  1. Point-by-point response to Comments and Suggestions for Authors from Reviewer 1

The manuscript describes a series of experiments carried out on various strains (clans) of Phaeodactylum tricornutum collected from an estuary near Blackpool (UK) and isolated from ecotype Pt2 (Plymouth, UK), and from Helgoland (Helgoland, Atlantic Ocean, North Sea, Germany ), pure tested morphotypes were isolated from these ecotypes. They were subjected to two long-term heat stresses (30°C and 37°C) to discover how an increase in temperature might affect morphological changes and the expression of photosynthesis-related genes in wild-type P. tricornutum and Ptezh mutants. Changes in cell morphology, photosynthesis process and H3K27me3 content were analyzed, which allowed to demonstrate the epigenetic control of this mark in the reactions of the diatom P. tricornutum to heat stress. The article is interesting and worth publishing with some corrections.

Comment 1: Line 25: In light of the statement that “in Arabidopsis thaliana, an opposite observation was reported. Growth under warm temperature (27°C) led to a genome-wide accumulation of H3K27me3 in this plant [16]. The epigenetic control of heat stress responses might thus be different in marine and terrestrial photosynthetic organisms.” (lines 486-489). The statement “Hence, we provided direct evidence of epigenetic control of the H3K27me3 mark in diatom responses to heat stress” should be modified. Most likely, the research results cannot be applied to all diatoms. This probably applies only to species living in the sea, and certainly applies to Phaeodactylum tricornutum.

We agree with Reviewer 1 that our findings might not be applicable to all diatoms. Therefore, we modified this sentence as followed: “Hence, we provided direct evidence of epigenetic control of the H3K27me3 mark in the response of Phaeodactylum tricornutum to heat stress.”

Comment 2: Line 74-76: Does shape change affect many (if not all) diatom species, or just Phaeodactylum tricornutum? I suggest clarifying this sentence.

P. tricornutum is the only known diatom to exist in several morphotypes. We added the following sentence to clarify this point: “P. tricornutum is found in coastal waters, including tidal areas where temperature can be highly variable due to changing water levels. Besides, temperature is known to impact P. tricornutum morphology [3]. This diatom being atypical in that it is the only species known to have three major morphotypes”.

Comment 3: Lines 70-75, The paragraph ending with the sentence "However, the impact of high temperature on marine diatoms remains mostly unknown, as well as the molecular players controlling morphotype conversion.", has nothing to do with the substantive content later in this paragraph (from the sentence: “The Polycomb Repressive Complex 2 (PRC2) is involved in cell differentiation in various eukaryotes [12].”) – I propose two paragraphs. I suggest moving the first one closer to the beginning of the "Introduction" section.

We agree with Reviewer 1 that these two sentences are not related enough. Therefore, as suggested, we moved the sentence " However, the impact of high temperatures on marine diatoms as well as the molecular players controlling morphotype conversion remain mostly unknown." in the first paragraph of the introduction.

Comment 4: Lines 83 and 99; 109, 135 and 136: Why is it sometimes written " Ptezh " and elsewhere " PtEZH"?

We followed a standard scientific convention used for naming genes and their corresponding proteins, as well as distinguishing between mutant and wild-type lines. Ptezh refer to the lines presenting a mutation in the PtEZH gene while PtEZH stands for the PtEZH protein. For protein designation, all letters are in upper case (PtEZH) while gene symbols are also in upper case but italicized and the designation is the same than the protein symbol (PtEZH). Mutant alleles are italicized and in lower case (Ptezh). We added “protein” at line 86 to define it when it first appeared in the text. For Ptezh, we mostly used Ptezh mutants unless it makes the syntax too cumbersome.

Comment 5: Lines 100-102: “We showed that the three morphotypes of P. tricornutum (fusiform, oval and triradiate) were affected in 101 their cell shape during heat stress …” - This has been shown previously [3], this study showed “only” , that this feature is lost in Ptezh mutants.

Indeed, in [3], cell shape was reported to change during heat stress for the fusiform Pt1 8.6 and oval Pt3 lines. However, in that study, no changes were reported in the morphotype abundance during a transition from cold (15°C) to heat (28°C) for the Pt8 ecotype (presenting a mixture of TM and FM cells). However, in the present study, we did show a change when triradiate lines (Pt8 and Pt16) were transferred from standard growth conditions (19°C) to heat stress (37°C). We showed that “both pure triradiate lines used in our study (Pt16-TM and Pt8-TM) showed conversion of TM to FM at 37°C (Fig. 4c and 6d) but not at 30°C (Fig. 4a and 6b)” as stated in lines 425-427. For this reason, we clarified the sentence as follows at lines 105-106: “We showed that the three morphotypes of P. tricornutum (fusiform, oval and triradiate) were affected in their cell shape during heat stress (after being grown under standard temperature conditions), a feature lost in Ptezh mutants.”

Comment 6: Line 119: According to Fig. 1a and 1b, only TM-M2.10 had a higher µmax value, and not as in the sentence "…. while TM-M2.11 displayed a higher one (Fig. 1b)”.

We agree with Reviewer 1 and changed the sentence at line 123-124 as follows: “FM-del6, FM-M2.3 and TM-M2.11 mutants presented a reduced µmax (Fig. 1a-b) while TM-M2.10 displayed a higher one (Fig. 1b)”.

Comment 7: Lines 225-226 and Figure 4: "The tropical line Pt9 did not show any changes in morphotype abundance in either MHS (Fig. 4a) ..." However, according to Figure 4a, between the 3rd and 4th day of line Pt9 exposure on temperature, the number of OM cells decreased by a similar amount as in the Pt1 8.6-FM morphotype.

We agree with Reviewer 1 and changed the sentence at line 234-235 as follows: “The tropical line Pt9 did not show any changes in morphotype abundance during the first two days in either MHS (Fig. 4a) or and in EHS (Fig. 4c).

Comment 8: Lines 438-439 and 461: "(Wan et al. 2023)" – not in the literature list.

We apologize for this omission. We added the reference to the literature list, and it now appears as [23] at lines 445 and 474 in the text.

Reviewer 2 Report

Comments and Suggestions for Authors

The manuscript presents a thorough investigation into the effects of moderate and elevated heat stresses on Phaeodactylum tricornutum, focusing on cell morphology, photosynthesis, and H3K27me3 epigenetic dynamics. By utilizing mutants deficient in PtEZH, responsible for H3K27me3 deposition, the study rigorously examines these responses in synchronized cell cultures under controlled conditions. The findings reveal PtEZH's pivotal role in regulating heat-induced morphological changes across different morphotypes (fusiform, oval, triradiate), as well as influencing gene expression related to photosynthesis and genome-wide H3K27me3 levels. This study significantly enhances our understanding of diatom adaptation mechanisms to heat stress. The experiment design and results are reliable and largely supported the main conclusions of this paper. However, strengthening clarity in experimental procedures and broadening the contextualization of results would further amplify the manuscript's impact in the field of marine biology and environmental science. 

1. The research explores a wide array of genes involved in photosynthesis, xanthophyll cycles, and heat shock, along with different morphotypes and temperature gradients. To enhance the clarity and accessibility of the manuscript, it would be advantageous to incorporate a summary figure that succinctly presents these genetic responses. This graphical representation would effectively synthesize the complex gene expression data, facilitating a clearer interpretation of how P. tricornutum responds molecularly to heat stress. Such an addition would not only streamline the presentation of results but also provide readers with a overview of the study's significant genetic findings within the broader context of diatom biology. 

2. The authors conducted an extensive qPCR analysis encompassing a wide array of genes, as detailed in Supplementary Table 1. It is noted that some genes were targeted with several primers. Are they genes with multiple copy numbers? Moreover, it would be beneficial to provide primer efficiencies for qPCRs for some key genes. Including this information would strengthen the reliability of the qPCR results and ensure that variations in gene expression are accurately interpreted. 

3. The manuscript would benefit from a more comprehensive description of the methodologies employed. Enhancing the detail regarding key experimental procedures and including thorough explanations of controls utilized would enhance the scientific rigor of the study and facilitate reproducibility by the research community. 

Minor points 

1. The figures (Figures 1, 2, and 3) in the manuscript depict three different treatments, but the significant differences are not clearly labeled between which two treatments. This omission could potentially confuse readers. It is recommended to use standard labeling methods. 

2. Lines 143-144: The original text is somewhat difficult to understand and reads awkwardly. Please revise to “the light level at which the increase in ETR stops being linear with light”. 

3. It would be valuable to discuss potential suggestions for future research based on the current findings. Highlight specific areas where further investigation could deepen understanding of PtEZH-mediated responses to environmental stressors, possibly involving other molecular pathways or epigenetic modifications.

Comments on the Quality of English Language

In my opinion, the english of this manuscript is of high standard. It effectively presents the results in a clear and coherent manner, making it easy to follow for readers. The expressions are precise, and the overall clarity enhances comprehension of the study's findings.

Author Response

Point-by-Point Response to Reviewer 2 comments:

  1. Summary: We would like to thanks both reviewers who provided detailed edits. We addressed them all. These comments greatly improved our manuscript entitled "H3K27me3 and EZH are involved in the control of the heat-stress-elicited morphological changes in diatoms" by Zarif et al. (ijms-3104800). We hope that these answers will satisfy both reviewers and the Editor and sufficiently improve our manuscript for publication in IJMS. Please find the detailed responses below and the corresponding corrections highlighted in red in the re-submitted files.

2. Questions for General Evaluation

Reviewer’s Evaluation

Response and Revisions

Does the introduction provide sufficient background and include all relevant references?

Yes

Is the research design appropriate?

Yes

Are the methods adequately described?

Yes

Addressed in Comments 3 and M1

Are the results clearly presented?

Can be improved

Addressed in Comments 2 and M1 and M2

Are the conclusions supported by the results?

Yes

  1. Point-by-point response to Comments and Suggestions for Authors from Reviewer 2

The manuscript presents a thorough investigation into the effects of moderate and elevated heat stresses on Phaeodactylum tricornutum, focusing on cell morphology, photosynthesis, and H3K27me3 epigenetic dynamics. By utilizing mutants deficient in PtEZH, responsible for H3K27me3 deposition, the study rigorously examines these responses in synchronized cell cultures under controlled conditions. The findings reveal PtEZH's pivotal role in regulating heat-induced morphological changes across different morphotypes (fusiform, oval, triradiate), as well as influencing gene expression related to photosynthesis and genome-wide H3K27me3 levels. This study significantly enhances our understanding of diatom adaptation mechanisms to heat stress. The experiment design and results are reliable and largely supported the main conclusions of this paper. However, strengthening clarity in experimental procedures and broadening the contextualization of results would further amplify the manuscript's impact in the field of marine biology and environmental science. 

Comment 1. The research explores a wide array of genes involved in photosynthesis, xanthophyll cycles, and heat shock, along with different morphotypes and temperature gradients. To enhance the clarity and accessibility of the manuscript, it would be advantageous to incorporate a summary figure that succinctly presents these genetic responses. This graphical representation would effectively synthesize the complex gene expression data, facilitating a clearer interpretation of how P. tricornutum responds molecularly to heat stress. Such an addition would not only streamline the presentation of results but also provide readers with a overview of the study's significant genetic findings within the broader context of diatom biology. 

We thank Reviewer 2 for this insightful suggestion, which indeed brings greater clarity to our paper and helps us provide a more comprehensive overview of our findings. We added a representation of the gene expression changes monitored by qPCR (i) for photosynthesis and xanthophyll cycles in both morphotypes (Fig. S3a) and for photosynthesis, xanthophyll cycles and heat shock in both heat stress conditions (Fig. S3b). In Fig. S3c, we synthesized the heat-elicited morphotype changes for each morphotype and genetic backgrounds (wild type and Ptezh mutants) in both heat stress conditions (MHS-30°C and EHS-37°C). We added citations for Fig. S3 in the manuscript (highlighted in red).

Comment 2. The authors conducted an extensive qPCR analysis encompassing a wide array of genes, as detailed in Supplementary Table 1. It is noted that some genes were targeted with several primers. Are they genes with multiple copy numbers? Moreover, it would be beneficial to provide primer efficiencies for qPCRs for some key genes. Including this information would strengthen the reliability of the qPCR results and ensure that variations in gene expression are accurately interpreted. 

Each gene analyzed by qPCR is targeted by only one couple of primers. Some genes have a close naming such as HSP20 and HSP20A, both being single genes not present in multiple copies. We conserved the naming of the Phatr3 annotation of P. tricornutum that is the latest one. Moreover, in P. tricornutum, the two genes Phatr3_J46657and Phatr3_J13358 (targeted by primers PETC-1_qPCR_F/R and PETC-2_qPCR_F/R, respectively) encode two different Rieske proteins also named petC proteins. We added the following sentence to clarify this at lines 184-185: “One should note that two genes were identified to encode Rieske proteins, petC1 (Phatr3_J46657) and petC2 (Phatr3_J13358) in P. tricornutum.” For each primer set, we added the qPCR primer efficiency in a column labelled “qPCR Efficiency” in the list of primers displayed in Table S1.

Comment 3. The manuscript would benefit from a more comprehensive description of the methodologies employed. Enhancing the detail regarding key experimental procedures and including thorough explanations of controls utilized would enhance the scientific rigor of the study and facilitate reproducibility by the research community. 

We provided more details in the Material and Methods section (highlighted in red).

Minor points 

Comment M1. The figures (Figures 1, 2, and 3) in the manuscript depict three different treatments, but the significant differences are not clearly labeled between which two treatments. This omission could potentially confuse readers. It is recommended to use standard labeling methods. 

We agree with Reviewer 2 that growth conditions were not clearly explained in Figures 1-3, Figures 2 and 3 corresponding to the same treatment. Thus, we provided more details in the legends of Figures 1-3 to describe the various treatments. Besides, significant differences were calculated between the wild type lines (either FM or TM) and the Ptezh mutants lines (FM-del6 and FM-M2.3 or TM-M2.10 and TM-M2.11). We modified Figures 1-3 to follow standard methods for statistically significant differences. The added details are highlighted in red.

Comment M2. Lines 143-144: The original text is somewhat difficult to understand and reads awkwardly. Please revise to “the light level at which the increase in ETR stops being linear with light”. 

To make it understandable, we rephrased the definition of Ek as followed: “the Ek coefficient, which is the light level at which ETR reaches a maximum and the relation between PAR irradiances and ETR stops to be linear” at lines 149-150.

Comment M3. It would be valuable to discuss potential suggestions for future research based on the current findings. Highlight specific areas where further investigation could deepen understanding of PtEZH-mediated responses to environmental stressors, possibly involving other molecular pathways or epigenetic modifications.

Indeed, including future perspectives based on the findings of our current study will be valuable to open avenues for future research in this field. We added the following paragraph in the discussion at lines 498-509: “Hence, this study opened new avenues regarding the investigation of heat stress responses that are under the control of PtEZH and H3K27me3 in P. tricornutum. As further perspectives, it will be relevant to investigate the genome-wide distribution of H3K27me3 and PtEZH in each morphotype (FM, OM or TM) upon heat stress. In parallel, a transcriptomic analysis will be pertinent to identify key regulators involved in the control of heat-elicited morphotype changes, based on changes in their expression concomitantly with their H3K27me3 levels. Besides, analyzing the cross talk between various epigenetic marks such as the transcription permissive and repressive ones (namely H3K4me3 and H3K36me3 vs. H3K9me3 and H3K27me3) as well as DNA methylation will be valuable to understand how epigenetics triggers heat-elicited morphotype changes in the model diatom P. tricornutum.”

Comments on the Quality of English Language

In my opinion, the english of this manuscript is of high standard. It effectively presents the results in a clear and coherent manner, making it easy to follow for readers. The expressions are precise, and the overall clarity enhances comprehension of the study's findings.

We are grateful to Reviewer 2 for this positive feedback. We are glad to hear that Reviewer 2 found the English in the manuscript to be of high standard and that the manuscript is clear and well presented.

Round 2

Reviewer 2 Report

Comments and Suggestions for Authors

I have carefully reviewed the revised manuscript and found that all of my concerns have been addressed.